# A Metalless and Fungicide-Free Material Against *Candida*: Glass-Loaded Hydrogels

**DOI:** 10.3390/pharmaceutics17070836

**Published:** 2025-06-26

**Authors:** Gabrielle Caroline Peiter, Elane da Silva Salvador, Fabián Ccahuana Ayma, Kádima Nayara Teixeira, Silvia Jaerger, Rafael A. Bini, Cleverson Busso, Rodrigo José de Oliveira, Ricardo Schneider

**Affiliations:** 1Group of Polymers and Nanostructures, Federal University of Technology-Paraná, Rua Cristo Rei, 19, Toledo 85902-490, Brazil; 2Physical Chemistry of Materials Group, Paraíba State University, Campina Grande 58429-500, Brazil; 3Immunology and Biochemistry of Pathological Processes Group, Federal University Paraná, Avenida Max Planck, 3796-Biopark, Toledo 85919-899, Brazil

**Keywords:** borophosphate glasses, glass-based hydrogel, fungicidal activity, *Candida* yeasts

## Abstract

**Background/Objectives:** We report the antifungal potential of transition metal-free borophosphate glass-loaded hydrogels (BGHs) with different phosphorus/boron molar ratios (P/B = 2, 1, and 0.5) against *Candida* species. Candida yeasts pose a significant health risk as they can cause infections, systemic diseases, and even potentially fatal complications in immunocompromised individuals. **Methods:** The antifungal activity of BGH was evaluated against *Candida albicans*, *Candida tropicalis*, *Candida krusei*, and *Candida glabrata* using kinetic growth analysis, the agar well diffusion method, the minimum inhibitory concentration, the minimum fungicidal concentration, and scanning electron microscopy. **Results:** All BGH formulations effectively inhibited yeast growth at various concentrations, with results comparable to commercial miconazole gel (CMG). Hydrogels with P/B ratios of 0.5 and 1 produced larger inhibition zones than CMG, except against *C. glabrata*. However, BGHs with a P/B ratio of 0.5 at 3% and 5% (*w*/*w*) demonstrated relevant antifungal activity, especially against *C. albicans* and *C. tropicalis*. **Conclusions:** These findings highlight the promising antifungal potential of borophosphate glass-based hydrogels, particularly those with high boron content. Their efficacy against multiple *Candida* species suggests they could serve as an alternative to conventional antifungal agents.

## 1. Introduction

Fungal infections caused by species of the genus *Candida* have become a global concern. Recently, the World Health Organization (WHO) considered *Candida albicans* a critical priority group in the development of new control drugs [1]. Globally, it is estimated that 400,000 people per year suffer from serious infections caused by *Candida* species. When the infection affects the blood system, the mortality rate is greater than 40% [2]. Candidosis is a term associated with skin, superficial, and deep tissue infections caused by *Candida* ssp. [3]. Yeast is considered commensal or, in certain situations, an opportunistic pathogen, colonizing mucosal surfaces of the oral cavity and digestive and genitourinary tracts [4]. The invasive process occurs after the disruption of gastrointestinal or skin barriers caused by injuries. In addition, intravascular catheter replacements in hospitals are also sources of invasive candidosis [5]. *Candida albicans* is the species with the highest infection prevalence, especially its phenotypic plasticity [6]. However, non-albicans species such as *Candida glabrata*, *Candida parapsilosis*, *Candida tropicalis*, and *Candida krusei* have also been found with variable frequency and resistance to the main antifungals [7,8], standing responsible for approximately 90% of candidosis [9].

Compared to the vast diversity and quantity of antibiotics available for bacteria, there are currently few clinically available antifungals [10,11]. Finding a drug that is selective, permeable to the fungal cell wall and membrane, and not toxic to the host is a major challenge [12]. Side effects such as bone marrow suppression and liver and kidney toxicity are the most common [13,14]. Miconazole, a synthetic imidazole antifungal, is widely used as a topical agent against yeasts and dermatophytes. Concentrations of 2 to 4% of the drug are normally used in the form of gels, solutions, or sprays against dermatitis caused by fungi, mainly *Candida albicans*. The drug generally has a minimum inhibitory concentration (MIC) of 1–10 μ g/mL for this recurrent opportunistic pathogen [15]. Although considered safe, studies have shown that the generation of ROS promoted by miconazole is not restricted only to fungal cells but also affects the host. The treatments described with miconazole are capable of inducing cytotoxic responses in keratinocyte cells by increasing ROS, presenting a negative correlation with the healing process of damage caused by the fungus [16]. Furthermore, studies have also proven the cytotoxic effect on cardiomyocytes and genotoxicity on germ and somatic cells of rats [17,18].

The development of new antifungal drugs, especially for new strains resistant to candidosis and with new mechanisms of action, is an emergency factor. Most of the new antifungals approved in the last 10 years are simply modifications of existing drug classes [19]. Compounds of inorganic origin with biocidal activity and the absence of microbial resistance have been the focus of new research with potential clinical application [20].

Glasses with antimicrobial activity have proven to be an innovative product against drug-resistant microorganisms. Many glasses are formulated or doped with metals such as zinc (Zn), copper (Cu), cobalt (Co), and silver (Ag) [21,22] to achieve antimicrobial activity. However, they are considered toxic heavy metals affecting cell growth, survival, and metabolism [23]. Furthermore, phosphate glasses can be obtained at relatively lower temperatures than ordinary silica-based glasses and their chemical resistance can be tuned with the addition of ions, e.g., aluminum [24]. In this way, soluble phosphate-based glasses can be exploited to develop dispersants and/or flocculants for clay such as Laponite^®^ [25]. Borophosphate-based glasses without heavy metals have been used in the form of hydrogels and have shown significant results against the bacteria *Staphylococcus aureus* and *Escherichia coli* [26].

In this work, we extended the use of transition metal-free borophosphate glasses as active agents in a hydrogel formulation as an antifungal against *Candida* species by changing the phosphorous/boron (P/B) ratio of the glasses. Optical density was employed to monitor the kinetic growth of *Candida albicans*, *Candida tropicalis*, *Candida krusei*, and *Candida glabrata*, while the antifungal performance was evaluated by the agar well diffusion method. Furthermore, the broth microdilution technique was used to determine the inhibitory and fungicidal concentration of the glass. As a tendency, the formulated hydrogels showed a superior activity than commercial miconazole gel, a reference drug, while the performance of the glass-formulated gels increases with the boron content and glass amount.

## 2. Experimental Section

### 2.1. Borophosphate Glass Synthesis and Glass-Loaded Hydrogel Formulation

Borophosphate glasses were synthesized using the melt-quenching technique, following a previously described methodology [26], with high-purity reagents (Sigma Aldrich, St. Louis, MO, USA). The molar ratio of KH_2_PO_4_/H_3_BO_3_ was set to 0.5, 1.0, or 2.0 to obtain the glasses 1P2B, 1P1B, and 2P1B, respectively. Two grams of raw materials were homogenized in an agate mortar. In a covered platinum/gold (95%/5%) crucible, the mixture was melted at 1050 °C for 1 h in a preheated resistive furnace. A graphite mold was used to apply the molten sample. After cooling, the samples were crushed in an agate mortar and stored in a desiccator. Borophosphate glass hydrogels (BGHs) were made using 1% (*w*/*w*) Carbopol^®^ (Sigma Aldrich, average 450,000) and 1% (*w*/*w*) glycerol (propane-1,2,3-triol). To obtain the BGHs, the 1P2B (3% and 5%), 1P1B (3% and 7%), or 2P1B (7% and 15%) was mixed with hydrogel and distilled water. A 0.01 M triethanolamine solution (Sigma Aldrich) was used to adjust pH to 6.0.

### 2.2. Analysis of the Yeast Growth Curves

The growth curve analysis was conducted for four *Candida* species. Initially, 100 μL of each hydrogel sample and positive control (CMG) were diluted in a 1:2 ratio with autoclaved distilled water. These diluted samples were subsequently distributed into 96-well microplates, which had been pre-inoculated with a suspension of the *Candida* species at a concentration of $10^{6}$ cells/mL, in duplicate. The plates were then incubated at 35 °C. Optical density (OD) measurements of the samples were taken at 530 nm at regular intervals over a 60 h period (from t = 0 to t = 60 h) using a microplate spectrophotometer (Thermo Scientific Multiskan Sky, Waltham, MA, USA). All analyses were performed in duplicate.

### 2.3. Antifungal Assays

The antifungal activity of the BGHs was evaluated by the minimum inhibitory concentration (MIC), minimum fungicidal concentration (MFC), and agar well diffusion method for four *Candida* species. The yeasts *Candida albicans* (ATCC 90028), *Candida tropicalis* (ATCC 750), *Candida krusei* (ATCC 6258), and *Candida glabrata* (ATCC 2001) were tested. MIC and MFC were assessed through serial concentration dilutions (*w*/*w*) of 2P1B (15% and 7%), 1P1B (7% and 3%), and 1P2B (5% and 3%) according to the guidelines of the Clinical and Laboratory Standards Institute (CLSI). The agar well diffusion method was performed according to previous protocols [27,28], with modifications, and the zones of inhibition were recorded. All the analyses were performed in triplicate.

The hydrogel without glass (HWG) and a 20 mg/g (2% *w*/*w*) commercial miconazole gel (CMG) were used as negative and positive controls, respectively. The results were analyzed according to the mean of the data ± standard deviation (SD) and compared using one-way ANOVA followed by Tukey’s post test to compare the activity of BGHs for each yeast species. GraphPad Prism 9.5.0. software was used. Significance of all the statistical tests was predetermined at p<0.05.

### 2.4. Scanning Electron Microscopy (SEM)

For the analysis by scanning electron microscopy (SEM), the contents of the wells from the antimicrobial activity test treated with the hydrogel (MIC, subminimum inhibitory concentrations (sub-MIC), controls: positive (CMG) and negative (HWG) (see Section 2.3) were collected and centrifuged (5000 rpm, 10 min), washed with PBS (0.1 M, pH 7.2), and fixed in a 2.5% (*v*/*v*) glutaraldehyde solution for 18 h at 4 °C, followed by another centrifugation and washing step. Post-fixation was performed using 1% osmium tetroxide (100 μL) for 1 h at 25 °C.

For slide preparation, circular glass coverslips (13 mm) were coated with a 0.1% poly-L-lysine solution (30–70 kDa, Sigma-Aldrich) for 1 h to promote cell adhesion. Then, 50 μL of the samples was deposited onto the coverslips and maintained at 50 °C until completely dry. The cells were subjected to a dehydration process using increasing concentrations of ethanol (50%, 70%, 80%, 90%, 95%, and 100%), with an exposure time of 10 min for each concentration, with the final step repeated twice.

After dehydration and critical point drying (CPD300 Critical Point Dryer, Leica EM, Wetzlar, Germany), the samples were analyzed using a VEGA3 scanning electron microscope (Tescan, Brno, Czech Republic) at magnifications of 5000× and 10,000×.

## 3. Results and Discussion

Figure 1 shows the growth curves of the *C. albicans* in the presence of the negative (HWG) and positive (CMG) controls and in the presence of the BGH 3% (*w*/*w*) 1P2B. Initially, a rapid growth (t = 9 h to t = 17 h), characterizing the exponential phase, was observed for *C. albicans* in the presence of the HWG (Figure 1a). Next, between t = 17 h and t = 22 h, there was a slowdown in yeast growth until reaching a threshold (stationary phase), followed by a decay (decline phase). The CMG (inset Figure 1b—red points) demonstrated antifungal activity against *C. albicans*, as expected. *C. tropicalis*, *C. krusei*, and *C. glabrata* exhibited growth curves similar to *C. albicans* in the presence of CMG and HWG (Appendix A). The green points in Figure 1b represent the 1P2B 3% (*w*/*w*), which exhibited potent activity against *C. albicans*. Growth was inhibited as early as the first hour of incubation, and the antifungal activity persisted until t = 60 h.

The growth curves of the *C. albicans*, *C. tropicalis*, *C. krusei*, and *C.glabrata* in the presence of the 1P2B (3% and 5% *w*/*w*), 1P1B (3% and 7% *w*/*w*), and 2P1B (7% and 15% *w*/*w*) are shown in Figure 2. All three glasses inhibited the growth of the *Candida* species. Antifungal activity was demonstrated within the first hour of incubation, which lasted until t = 60 h.

Regarding the agar well diffusion method, the zone of inhibition obtained for the 1P2B (3% and 5% *w*/*w*), 1P1B (3% and 7% *w*/*w*), and 2P1B (7% and 15% *w*/*w*) compared with CMG are summarized in Table 1. The inhibition of growth in response to the BGHs was different for the four *Candida* species, and the inhibition of each species also varied for the different BGHs. The growth of *C. albicans* was inhibited by all BGHs, with this inhibition being greater than that caused by CMG, except for 2P1B 7% (*w*/*w*). 1P2B 5% (*w*/*w*) was the most effective in inhibiting the growth of *C. albicans*, and its activity was compared to the other BGHs (2P1B and 1P1B) at higher concentrations (15% and 7% (*w*/*w*), respectively). The growth of *C. tropicalis* was inhibited by all BGHs; however, growth inhibition by CMG was greater than that by the BGHs, except for 1P2B at 5% (*w*/*w*), as shown in Figure 3.

The boron-rich BGH at the highest concentration (5% *w*/*w*) demonstrated better antifungal activity against *C. albicans* and *C. tropicalis*. These species are the leading cause of candidosis and are common in patients with neutropenia and malignancy, respectively. Considering the 1P2B, a boron-rich BGH, even its decrease in the hygroscopicity increased the bond strength and cross-linking of the network by the formation of P−O−B bonds [24]. This fact can explain the better antifungal activity against *C. albicans* and *C. tropicalis*. Studies have shown that *Candida* species resistant to antifungals have developed due to the fungus’s genomic plasticity. When an azole-type antifungal is administered, the number of efflux cells in the *C. albicans* cell increases. These pumps, responsible for intracellular transport, work to prevent intracellular drug accumulation in order to avoid toxic levels capable of killing the cell. *C. tropicalis* is resistant to azoles and has low resistance to echinocandins (caspofungin) [12,29].

The highest average inhibition zones were observed for *C. krusei*. All BGHs inhibited the growth of the yeast compared to CMG. 1P2B at 5% (*w*/*w*) and 1P1B at 7% (*w*/*w*) were the most efficient in inhibiting the growth of *C. krusei*, and although no statistical difference was observed between the two glasses, both were more effective than the other glasses. Regarding *C. glabrata*, although CMG was more effective in inhibiting growth, all BGHs also inhibited yeast growth. The most effective BGHs were 2P1B 15% (*w*/*w*) and 1P1B 7% (*w*/*w*), with the growth inhibition caused by the latter being greater than that of 2P1B (Figure 3).

The largest zones of inhibition for *C. krusei* and *C. glabrata* were in the presence of the 1P1B (a boron-intermediate-BGH) at highest concentration (7% *w*/*w*), 35.6 mm and 27.6 mm, respectively. As observed in Table 1 and Figure 3, the boron-poor BGH (2P1B) was the least effective in inhibiting the growth of all *Candida* species. These results suggest that the *Candida* species analyzed in this study showed low resistance to BGHs with a lower boron molar ratio, such as 2P1B. Considering that the increase of the P/B ratio enhances the number of P-O-B bonds in the glass and boron units [30], the formation of hydrated borophosphates [31] and the boron-containing bonds in the glass structure are associated with an increase in antifungal activity. The antifungal mechanism is probably related to the disruption of carbohydrate metabolism as the proposed mechanism when using boron-containing compounds [32]. At the same time, the inhibition of oxidative metabolism is reported as the antifungal mechanism for pure boric acid against *Candida albicans* [33].

Table 2 shows the MIC and MFC values of the 2P1B (15% and 7% (*w*/*w*)), 1P1B (7% and 3% (*w*/*w*)), and 1P2B (5% and 3% (*w*/*w*)) for the four *Candida* species. The data indicate that CMG showed the lowest MIC and MFC values for the four *Candida* species compared to the BGHs. In relation to the MIC values of the BGHs, it was observed that 1P2B (3% and 5% (*w*/*w*)) (boron-rich BGH) shows fungistatic activity at lower concentrations for the four *Candida* species, with the lowest concentrations observed for *C. glabrata*. 1P2B was the most effective in inhibiting the growth of both pathogenic species *C. albicans* and *C. tropicalis*, corroborating the results of the agar well diffusion test. This result emphasizes the tendency towards multidrug resistance to traditional therapy with azoles and polyenes observed in more pathogenic species [12]. Regarding the fungicidal activity of the BGHs, the MFC results followed the trend of fungicidal activity—1P2B (3% and 5% (*w*/*w*)) was more effective in killing the four yeast species, followed by 1P1B (3% (*w*/*w*)). The BGHs exhibited better fungicidal activity against *C. glabrata* and *C. krusei* (lower MFC values). *C. krusei* infections are characterized by their high mortality rate (40–58%) and poor response to standard antifungal therapies. Accurate diagnosis and appropriate treatment are essential for controlling *C. krusei* infections, and choosing the appropriate antifungal can be challenging due to its resistance to some common antifungal medications. Due to the characteristics of the cell wall, there is a relationship with the relevance of this structure during interaction with the host and because it is the target of some antifungal drugs [12,29]. *C. glabrata* has been described in the literature as an emerging pathogen and is observed with a higher incidence in adults than in children and neonates. It is considered the second most frequently isolated *Candida* infection and is responsible for hospital resistance. Multidrug resistance of *C. glabrata* resulted from the increased use of azoles and echinocandins, which caused selective pressure on its cells [12,29]. In summary, all the BGHs exhibited both fungistatic and fungicidal activity against the tested *Candida* species. In particular, the BGHs showed an impressive performance on *C. krusei* and *C. glabrata* when compared to the positive control (CMG). The results indicate a promising approach for the development of borophosphate-based glass materials with antifungal activity for *Candida* species. Thus, considering the formation of an inhibition zone on the plates, monitored by agar wells, the borophosphate glasses were considered antifungal (Appendix A).

In contrast to prior studies highlighting the fungistatic and fungicidal properties of compounds like clotrimazole [34], natural compounds [35], and synthetic molecules with selenium [36], zinc oxide [37], and silver nanoparticles [38], in this study, the BGHs demonstrated antifungal activity either comparable or superior to commercial antifungal medicine. Furthermore, many organic compounds of plant or synthetic origin suffer from unfavorable physicochemical properties such as poor water solubility and high Log D values, potentially compromising therapeutic efficacy. The borophosphate composition used not only exhibited potent antifungal effects but also enabled complete dissolution in aqueous media. This dissolution characteristic holds promise for enhancing drug bioavailability at infection sites, circumventing the need for excessive polymer or surfactant usage to improve bioavailability, as often required with organic molecules or nanoparticles.

Figure 4 shows the scanning electron micrographs of *C. albicans* cells treated with BGH hydrogel (2P1B 15%, Table 1) for 24 h and compared to the minimum inhibitory concentration (MIC), minimum inhibitory concentrations (sub-MICs), and hydrogel without glass (HWG). The untreated cells, the yeast, appeared as rounded cells with a regular shape; smooth, homogeneous, and intact cell walls; and characteristic budding, as expected for planktonic cells, as observed in Figure 4A,E.

In the cells treated with hydrogel at the MIC (Figure 4B,F), no cellular debris or cells were present, indicating complete growth inhibition and demonstrating the effectiveness of the treatment compared to the sub-MIC condition (Figure 4C,G). The latter exhibited morphological alterations, including surface damage such as holes in the cell wall, irregular shape, and leakage of intracellular material. The images also revealed cells undergoing disintegration, along with fragments of already disintegrated cells. In the negative control group (hydrogel without glass—HWG), *C. albicans* cells attached to the gel structure without morphological alterations, maintaining their original shape and integrity, as observed in Figure 4D,H.

## 4. Conclusions

The development of novel antifungal therapies to combat drug-resistant *Candida* species is increasingly urgent. Our investigation underscores the significant antifungal potential of borophosphate-based glass materials, as evidenced by in vitro assays conducted against *Candida* species. The BGHs showed both fungistatic and fungicidal activity against *Candida* species, including *C. albicans*—the most prevalent in cases of candidiasis. Specifically, the boron-rich BGH (1P2B) displayed heightened activity against *Candida* species, followed by 1P1B (3% and 7% (*w*/*w*)). Conversely, boron-poor BGH (2P1B) exhibited lower antifungal activity against yeast species. Due to its strong antifungal properties, the investigated formulation has potential for various medical applications. This advancement is particularly crucial, considering the relatively limited therapeutic repertoire for antifungal agents compared to the plethora of antibiotics available for bacterial infections.

## Figures and Tables

**Figure 1 pharmaceutics-17-00836-f001:**
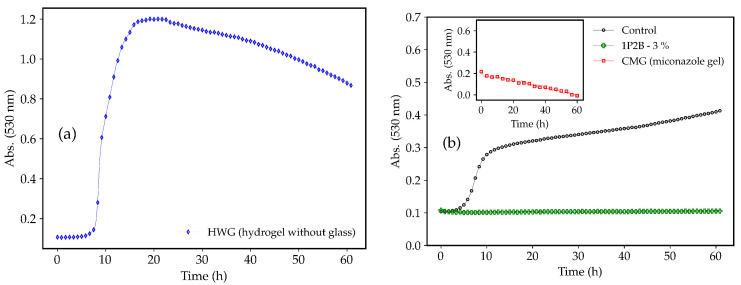
Growth curves of *C. albicans* in hydrogel without glass (HWG) (**a**), commercial miconazole gel (CMG), and borophosphate glass-loaded hydrogel (BGH) 3% (*w*/*w*) 1P2B (**b**). Control = *C. albicans* in RPMI 1640 medium only.

**Figure 2 pharmaceutics-17-00836-f002:**
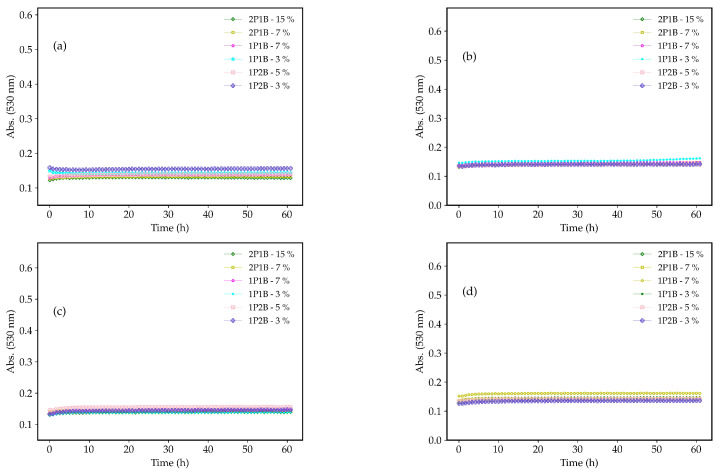
Growth curves of the (**a**) *C. albicans*, (**b**) *C. tropicalis*, (**c**) *C. krusei*, and (**d**) *C. glabrata* in the presence of the 1P2B (3% and 5% *w*/*w*), 1P1B (3% and 7% *w*/*w*), and 2P1B (7% and 15% *w*/*w*). Experiments performed in duplicate.

**Figure 3 pharmaceutics-17-00836-f003:**
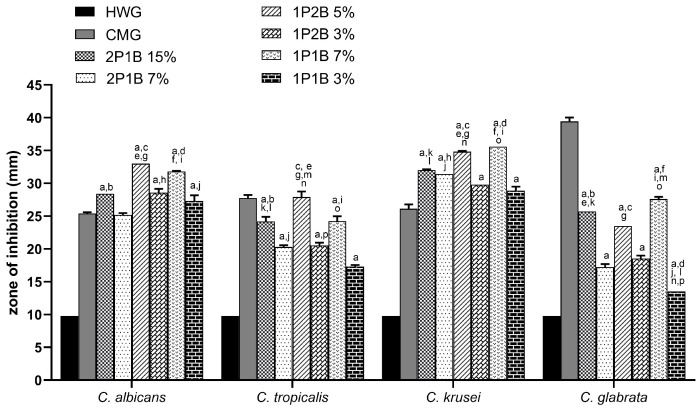
Mean zones of inhibition for *Candida* species cultures in association with borophosphate glass-loaded hydrogels (BGHs) 2P1B (15% and 7%),1P2B (5% and 3%), and 1P1B (7% and 3%); hydrogel without glass (HWG); and commercial miconazole gel (CMG). The symbols represent the statistically significant difference (p⩽0.05) among (a) miconazole 2% vs. all; (b) 2P1B 15% vs. 2P1B 7%; (c) 1P2B 5% vs. 1P2B 3%; (d) 1P1B 7% vs. 1P1B 3%; (e) 2P1B 15% vs. 1P2B 5%; (f) 2P1B 15% vs. 1P1B 7%; (g) 2P1B 7% vs. 1P2B 5%; (h) 2P1B 7% vs. 1P2B 3%; (i) 2P1B 7% vs. 1P1B 7%; (j) 2P1B 7% vs. 1P1B 3%; (k) 2P1B 15% vs. 1P2B 3%; (l) 2P1B 15% vs. 1P1B 3%; (m) 1P2B 5% vs. 1P1B 7%; (n) 1P2B 5% vs. 1P1B 3%; (o) 1P2B 3% vs. 1P1B 7%; and (p) 1P2B 3% vs. 1P1B 3%.

**Figure 4 pharmaceutics-17-00836-f004:**
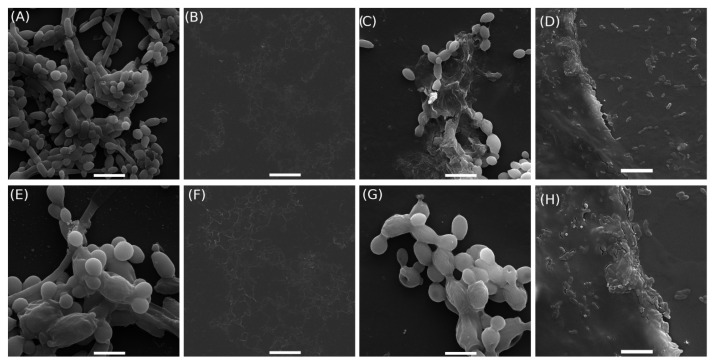
Scanning electron microscopy of cells: *Candida Albicans* cell culture RMPI (**A**,**E**); minimum inhibitory concentration (MIC, borophosphate glass concentration 2P1B 15% (Table 1)) (**B**,**F**); subminimum inhibitory concentrations (sub-MICs, borophosphate glass concentration 2P1B 4.6 mg/mL) (**C**,**G**); and negative control (hydrogel without glass (HWG)) (**D**,**H**). Images (**A**–**D**) scale bar: 10 μm. Images (**E**–**H**) scale bar: 5 μm.

**Table 1 pharmaceutics-17-00836-t001:** Zone of inhibition (diameter) of the *Candida albicans*, *Candida tropicalis*, *Candida krusei*, and *Candida glabrata* in the presence of BGHs with different concentrations of borophosphate glass.

Yeast	Agent^ (a)^	CMG ^(b)^	Borophosphate Glass Concentration^ (c)^			
		**2%**	**15%**	**7%**	**5%**	**3%**
*C. albicans*	CMG	25.4±0.2	-	-	-	-
	2P1B	-	28.4±0.0	25.2±0.3	-	-
	1P2B	-	-	-	33.0±0.0	28.6±0.6
	1P1B	-	-	31.8±0.1	-	27.3±0.8
*C. tropicalis*	CMG	27.7±0.5	-	-	-	
	2P1B	-	24.2±0.7	20.3±0.3	-	-
	1P2B	-	-	-	27.9±0.9	20.5±0.4
	1P1B	-	-	24.2±0.8	-	17.3±0.3
*C. krusei*	CMG	26.1±0.5	-	-	-	-
	2P1B	-	32.0±0.1	31.4±0.0	-	-
	1P2B	-	-	-	34.8±0.1	29.8±0.0
	1P1B	-	-	35.6±0.0	-	28.9±0.6
*C. glabrata*	CMG	39.4±0.6	-	-	-	-
	2P1B	-	25.7±0.1	17.2±0.4	-	-
	1P2B	-	-	-	23.5±0.0	18.5±0.5
	1P1B	-	-	27.6±0.3	-	13.5±0.0

^(a)^ Zone of inhibition is represented in millimeters (mm) ± % standard deviation (SD); ^(b)^ commercial miconazole
gel (CMG); ^(c)^ concentration in weight base.

**Table 2 pharmaceutics-17-00836-t002:** MIC and MFC analyses of different concentrations of hydrogels containing borophosphate glass.

Agent	*C. albicans*		*C. tropicalis*		*C. krusei*		*C. grablata*	
	**MIC ^a^**	**MFC ^b^**	**MIC**	**MFC**	**MIC**	**MFC**	**MIC ^b^**	**MFC**
CMG 2% ^c^	<0.04	<0.04	<0.04	<0.04	<0.04	<0.04	<0.04	<0.04
2P1B 7%	8.7	17.5	8.7	17.0	2.1	4.3	<0.1	<0.1
2P1B 15%	9.3	18.7	9.3	18.7	2.3	4.6	<0.2	<0.1
1P1B 3%	3.7	7.5	7.5	>7.5	0.9	1.8	<0.05	<0.05
1P1B 7%	8.7	17.5	4.3	17.5	2.1	4.3	<0.1	<0.1
1P2B 3%	1.8	7.5	3.7	7.5	0.9	1.8	<0.05	<0.05
1P2B 5%	1.5	12.5	3.1	12.5	0.7	1.5	<0.09	<0.09

^a^ mg/mL; ^b^ the symbol > indicates that the value is higher than the highest initial concentration evaluated or
< indicates that the value is probably lower than concentration observed; ^c^ commercial miconazole gel (CMG).

## Data Availability

The data that support the findings of this study are available from the corresponding author upon reasonable request.

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
