# Peer review of "A Metalless and Fungicide-Free Material Against Candida: Glass-Loaded Hydrogels"

_pharmaceutics, 2025, doi:10.3390/pharmaceutics17070836_

Round 1
Reviewer 1 Report
Comments and Suggestions for Authors
This manuscript addresses a topic — the development of fungicide-free, metal-free hydrogels loaded with borophosphate glasses (BGH) as an alternative antifungal strategy against Candida spp. The work is of potential interest to the antimicrobial materials community and offers a relevant comparison with commercial miconazole gel. The experimental approach is methodologically sound, and the antifungal activity is evaluated using multiple complementary techniques. However, while the study presents promising findings, there are critical aspects that require clarification or improvement.
- I recommend the authors revise the introduction section (prior to line 60) to address the following issues, including define the limitations of existing metal-containing or fungicide-based materials, and articulate the novelty of “metal-free and fungicide-free” materials.
-
Regarding the antifungal mechanism, the authors only briefly mention the P–O–B bonds and the network cross-linking strength, but do not provide an in-depth discussion.
- The arrangement of data and unit annotations in Table 1 appear disorganized.
- Although ANOVA and Tukey's post hoc test are mentioned, there is a lack of detailed reporting on statistical significance, such as p-values.
- Line246,The conclusion states that "the next step will involve animal model experiments," but it does not specify which model will be used or what evaluation criteria will be applied. Moreover, this information is not appropriate for the conclusion section.
Author Response
Dear Reviewer,
First of all, we would like to thank you and the referees for their valuable comments and suggestions to improve the quality of the manuscript. It is very important to emphasize that, in this study, we presented an effective and new approach to obtaining glass-loaded hydrogels. The hydrogel effectively inhibited Candida growth with performance comparable to commercial miconazole gel. Notably, hydrogels with P/B ratios of 0.5 and 1 produced larger inhibition zones, highlighting their promising potential as alternative antifungal agents.
The reviewers indicated the following clarifications and changes, as shown below in Italic. Our responses have been marked in red and amendments/additions to the revised manuscript/SI in blue.
In the thoroughly revised manuscript, we have addressed all requests of the editorial office and referees' comments. Following are individual suggestions/questions and their respective response:
Reviewer 1#
This manuscript addresses a topic — the development of fungicide-free, metal-free hydrogels loaded with borophosphate glasses (BGH) as an alternative antifungal strategy against Candida spp. The work is of potential interest to the antimicrobial materials community and offers a relevant comparison with commercial miconazole gel. The experimental approach is methodologically sound, and the antifungal activity is evaluated using multiple complementary techniques. However, while the study presents promising findings, there are critical aspects that require clarification or improvement.
1) I recommend the authors revise the introduction section (prior to line 60) to address the following issues, including define the limitations of existing metal-containing or fungicide-based materials, and articulate the novelty of “metal-free and fungicide-free” materials.
Our answer:
The Introduction section was revised.
We added the following sentence (lines 58-68) with three new references (numbers [23], [24], and [25]) in the revised manuscript.
However, they are considered toxic heavy metals affecting cell growth, survival and metabolism [23]. Furthermore, phosphate glasses can be obtained at relatively lower temperatures than ordinary silica-based glasses and tuned their chemical resistance by addition of ions, e.g. aluminium [24]. In this way, soluble phosphate-based glasses can be exploited to develop dispersants and/or flocculants for clay as Laponite® [25].
2) Regarding the antifungal mechanism, the authors only briefly mention the P–O–B bonds and the network cross-linking strength, but do not provide an in-depth discussion.
Our answer:
We thank the referee for the comment. Boron is a non-essential element for fungi. The disruption of carbohydrate metabolism is reported as the major mechanism when using boron-containing compounds [DOI: 10.1016/j.jtemb.2021.126714]. Specifically, for Candida Albicans the inhibition of oxidative metabolism is reported as the antifungal mechanism for boric acid [ DOI: 10.1093/jac/dkn486]. Considering that the increase of the P/B ratio enhances the number of P-O-B bonds in the glass and boron units [DOI: 10.1021/acs.jpcb.7b01563] the formation of hydrated borophosphates [DOI: 10.1016/j.nocx.2023.100181] and the boron-containing bonds in the glass structure are associated with increase of antifungal activity.
Accepting the referee's observation and to clarify the point we have rewrite the sentence from :
”...This fact can be explained by the bond strength and cross-linking of the network of the P-O-B bonds is lower compared to the 1P2B (boron-rich BGH).”
To
Considering that the increase of the P/B ratio enhances the number of P-O-B bonds in the glass and boron units [30], the formation of hydrated borophosphates [31] and the boron-containing bonds in the glass structure are associated with increase of antifungal activity. The antifungal mechanism is probably related to the disruption of carbohydrate metabolism as the proposed mechanism when using boron-containing compounds [32]. At the same time, the inhibition of oxidative metabolism is reported as the antifungal mechanism for pure boric acid against Candida Albicans [33].
The following sentences were add
30. Anastasopoulou, M.; Vasilopoulos, K.C.; Anagnostopoulos, D.; Koutselas, I.; Papayannis, D.K.; Karakassides, M.A. Structural and Theoretical Study of Strontium Borophosphate Glasses Using Raman Spectroscopy and ab Initio Molecular Orbital Method. The Journal of Physical Chemistry B 2017, 121, 4610–4619. https://doi.org/10.1021/acs.jpcb.7b01563.
31. Freudenberger, P.T.; Blatt, R.L.; Brow, R.K. Dissolution rates of borophosphate glasses in deionized water and in simulated body fluid. Journal of Non-Crystalline Solids: X 2023, 18, 100181. https://doi.org/10.1016/j.nocx.2023.100181.
32. Estevez-Fregoso, E.; Farfán-García, E.D.; García-Coronel, I.H.; Martínez-Herrera, E.; Alatorre, A.; Scorei, R.I.; Soriano-Ursúa, M.A. Effects of boron-containing compounds in the fungal kingdom. Journal of Trace Elements in Medicine and Biology 2021, 65, 126714. https://doi.org/10.1016/j.jtemb.2021.126714.
33. De Seta, F.; Schmidt, M.; Vu, B.; Essmann, M.; Larsen, B. Antifungal mechanisms supporting boric acid therapy of Candida vaginitis. Journal of Antimicrobial Chemotherapy 2008, 63, 325–336. https://doi.org/10.1093/jac/dkn486.
3) The arrangement of data and unit annotations in Table 1 appear disorganized.
Our answer:
Table 1 was modified by adding a trace in the cells without data. Additionally the superscript text was the position changed for better visualization.
4) Although ANOVA and Tukey's post hoc test are mentioned, there is a lack of detailed reporting on statistical significance, such as p-values.
Our answer:
We thank the referee for the comment.
The following complement was inserted in the line 111-112:
Significance of all the statistical tests was predetermined at p <0 .05.
5) Line246,The conclusion states that "the next step will involve animal model experiments," but it does not specify which model will be used or what evaluation criteria will be applied. Moreover, this information is not appropriate for the conclusion section.
Our answer:
The statement was removed.
Reviewer 2 Report
Comments and Suggestions for Authors
Make corrections in the file

Need some corrections
Author Response
Dear Reviewer,
First of all, we would like to thank you and the referees for their valuable comments and suggestions to improve the quality of the manuscript. It is very important to emphasize that, in this study, we presented an effective and new approach to obtaining glass-loaded hydrogels. The hydrogel effectively inhibited Candida growth with performance comparable to commercial miconazole gel. Notably, hydrogels with P/B ratios of 0.5 and 1 produced larger inhibition zones, highlighting their promising potential as alternative antifungal agents.
The reviewers indicated the following clarifications and changes, as shown below in Italic. Our responses have been marked in red and amendments/additions to the revised manuscript/SI in blue.
In the thoroughly revised manuscript, we have addressed all requests of the editorial office and referees' comments. Following are individual suggestions/questions and their respective response:
â€Our answer:‬
The authors thanks the referee's observations. The manuscript was revised.

Reviewer 3 Report
Comments and Suggestions for Authors
Manuscript "A metalless and fungicide-free material against Candida: glass-loaded hydrogels", by Gabrielle Caroline Peiter, Elane da Silva Salvador, Fabián Ccahuana Ayma, Kádima Nayara Teixeira, Silvia Jaerger, Rafael A. Bini, Cleverson Busso, Rodrigo José de Oliveira, and Ricardo Schneider.
The manuscript is very concise, without superfluous parts, with very effective results. Man's endless struggle with pathogenic microorganisms has been given another possibility and another chance in this manuscript. The authors presented the antifungal potential of transition metal-free borophosphate glass-loaded hydrogels (BGH) with different phosphorus/boron molar ratios against Candida species.
The idea is not very new, it was developed by other authors (Jaqueline Saracini et all, 2023), but the authors of this manuscript continued the research of such gels on Candida species. Therefore, I believe that these results will be of interest to other researchers dealing with similar topics.
Also, I will be free to suggest to the authors in their future research (not required for this manuscript) to perform cytotoxicity on healthy cell cultures for their borophosphate glass-loaded hydrogels (BGH) formulations. This would show whether such formulations have the potential for further development for human use.
Some corrections are needed in the manuscript. The authors did not format the references in their manuscript according to the guidelines for authors. I ask the authors to format the cited references according to "pharmaceutics-template.dot", i.e. Abbreviated Journal Name (italic) Year (bold), Volume (italic), etc.
I consider that the manuscript should be published in the journal "Pharmaceutics" after minor corrections.
Author Response
Dear Reviewer,
First of all, we would like to thank you and the referees for their valuable comments and suggestions to improve the quality of the manuscript. It is very important to emphasize that, in this study, we presented an effective and new approach to obtaining glass-loaded hydrogels. The hydrogel effectively inhibited Candida growth with performance comparable to commercial miconazole gel. Notably, hydrogels with P/B ratios of 0.5 and 1 produced larger inhibition zones, highlighting their promising potential as alternative antifungal agents.
The reviewers indicated the following clarifications and changes, as shown below in Italic. Our responses have been marked in red and amendments/additions to the revised manuscript/SI in blue.
In the thoroughly revised manuscript, we have addressed all requests of the editorial office and referees' comments. Following are individual suggestions/questions and their respective response:
Manuscript "A metalless and fungicide-free material against Candida: glass-loaded hydrogels", by Gabrielle Caroline Peiter, Elane da Silva Salvador, Fabián Ccahuana Ayma, Kádima Nayara Teixeira, Silvia Jaerger, Rafael A. Bini, Cleverson Busso, Rodrigo José de Oliveira, and Ricardo Schneider.
The manuscript is very concise, without superfluous parts, with very effective results. Man's endless struggle with pathogenic microorganisms has been given another possibility and another chance in this manuscript. The authors presented the antifungal potential of transition metal-free borophosphate glass-loaded hydrogels (BGH) with different phosphorus/boron molar ratios against Candida species.
The idea is not very new, it was developed by other authors (Jaqueline Saracini et all, 2023), but the authors of this manuscript continued the research of such gels on Candida species. Therefore, I believe that these results will be of interest to other researchers dealing with similar topics.
Also, I will be free to suggest to the authors in their future research (not required for this manuscript) to perform cytotoxicity on healthy cell cultures for their borophosphate glass-loaded hydrogels (BGH) formulations. This would show whether such formulations have the potential for further development for human use.
Some corrections are needed in the manuscript. The authors did not format the references in their manuscript according to the guidelines for authors. I ask the authors to format the cited references according to "pharmaceutics-template.dot", i.e. Abbreviated Journal Name (italic) Year (bold), Volume (italic), etc.
I consider that the manuscript should be published in the journal "Pharmaceutics" after minor corrections.
Our answer:
The authors would like to thank the reviewer comments and the positive consideration for publication. The referee’s suggestions will be considered in the next work. The author's clarify that the manuscript was written using LaTeX programming language using the "documentclass" option "pharmaceutics". Thus, further modifications in the text format, references and overall template can be done by the editorial office changing the options in style files of the MDPI LaTeX template.

Reviewer 4 Report
Comments and Suggestions for Authors
1. Instead of increasing the boron content, why did the authors use the 1P2B ratio?
2. Are there any plans to conduct cell cytotoxicity testing of BGH in animal models?
3. Did the authors consider the variation in the potency of BGH for different strains of Candida species?
4. The aqueous solubility of BGH is considered an advantage in this work, but will this affect its stability and shelf life in the long term? Have you conducted any stability testing? Or do you plan to do any in the future?
5. Why was BGH less effective against C. glabrata compared to CMG? The paper should address this issue in more detail. Can the authors highlight possible reasons why this happened?
Author Response
Dear Reviewer,
First of all, we would like to thank you and the referees for their valuable comments and suggestions to improve the quality of the manuscript. It is very important to emphasize that, in this study, we presented an effective and new approach to obtaining glass-loaded hydrogels. The hydrogel effectively inhibited Candida growth with performance comparable to commercial miconazole gel. Notably, hydrogels with P/B ratios of 0.5 and 1 produced larger inhibition zones, highlighting their promising potential as alternative antifungal agents.
The reviewers indicated the following clarifications and changes, as shown below in Italic. Our responses have been marked in red and amendments/additions to the revised manuscript/SI in blue.
In the thoroughly revised manuscript, we have addressed all requests of the editorial office and referees' comments. Following are individual suggestions/questions and their respective response:
- Instead of increasing the boron content, why did the authors use the 1P2B ratio?
Our answer:
During the glass synthesis, the ratio of the elements plays a significant role (for example, DOI: 10.1016/j.jnoncrysol.2017.11.034 and DOI: 10.1016/j.ceramint.2019.06.194). For example, the change in the amount of just one element can induce crystallization during quenching. Additionally, we can achieve a region (in the phase diagram) where the glass material cannot be obtained and/or we can change the properties of the glass matrix (DOI : 10.1016/j.jnoncrysol.2017.12.021).
- Are there any plans to conduct cell cytotoxicity testing of BGH in animal models?
Our answer:
Currently, through academic partners, the toxicity of pure polyphosphate and borophosphate glasses is being evaluated with rats of the species Rattus norvegicus, Wistar strain, aged 90 to 120 days, of both sexes. The parameters for carrying out this study will be based on the OECD (Organization for Economic Cooperation and Development) Guidelines for testing chemical products, specifically the “Repeated Dose 28-Day Oral Toxicity Study in Rodents”. Thus, these results will extend the current uses, where polyphosphate-based materials are used as biocidal materials for controlling bacteria in poultry farms.
- Did the authors consider the variation in the potency of BGH for different strains of Candida species?
Our answer:
The team considers the present work positive, encouraging us to conduct further studies using BGH against Candida strains. The variation in sensitivity to BGH is observed in Table 1. Different species of Candida present high genetic variability (DOI 10.1186/s12879-015-0793-3) allowing the emergence of new strains with variations in the composition of the cell structure, which also represents different sensitivity to antifungals.
Currently, the work is focused on evaluating microorganisms to show the broad spectrum activity. The glasses showed activity against bacteria P. aeruginosa, S. aureus, E. coli, (previous work DOI: 10.1016/j.ijpharm.2023.123323) and K. pneumoniae, L. Monocytogenes. Considering the reviewer question, a more detailed study is being conducted on Salmonella serovars: S. typhimurium, S. enteritidis, S. derby, S. senftenberg, S. choleraesuis, S. Heidelberg and fungi: Macrophamina phaseolina, Aspergillus niger, and Sclerotinia sclerotiorum.
- The aqueous solubility of BGH is considered an advantage in this work, but will this affect its stability and shelf life in the long term? Have you conducted any stability testing? Or do you plan to do any in the future?
Our answer:
Taking in account the pH stability of similar polyelectrolytes, like polyphosphates, it is expected that BGH could suffer slow hydrolysis under very acid pH. The pH of BGH was adjusted to pH=6 (section 2.1, lines 82-83 of the original manuscript). Polyphosphates are quite stable at neutral pH (pH 6.0 to 6.8) even at temperatures of 100°C (DOI: 10.3168/jds.2017-12764 and DOI: 10.1002/jpln.200420494), but relatively fast at pH close to zero (DOI: 10.1366/0003702981944535)
- Why was BGH less effective against C. glabrata compared to CMG? The paper should address this issue in more detail. Can the authors highlight possible reasons why this happened?
Our answer:
Miconazole is a drug with low water solubility and inhibits the ergosterol biosynthesis in the fungal cell membrane. The consequences are changes in permeability and also in cell division. BGH, in contrast, is water soluble, and its mechanism of action is not yet fully understood, however, sub-MIC concentrations reveal holes in the cell wall, irregular shape and leakage of intracellular material, revealing the action on the fungal cell wall (Figure 4). Candida glabrata, unlike other Candidas, has a bilayer cell wall with the outer part full of mannoproteins, and the amino acids have 65% identity with Saccharomyces cerevisiae (https://doi.org/10.1111/j.1742-4658.2012.08564.x). Furthermore, it is the only one among other pathogenic Candida species that does not have the ability to alter the cell wall to form pseudohyphae (doi: 10.1128/EC.00284-08). This significant difference in the composition and organization of the cell wall of C. glabrata may reduce the efficiency of the BCG mechanism of action.
Round 2
Reviewer 1 Report
Comments and Suggestions for Authors
The authors have addressed all my previous concerns satisfactorily. The revisions are appropriate, and I have no further issues.